# Impact of Coal-fired Power Plant Emissions on Children’s Health: A Systematic Review of the Epidemiological Literature

**DOI:** 10.3390/ijerph16112008

**Published:** 2019-06-05

**Authors:** Eric Amster, Clara Lew Levy

**Affiliations:** 1Department of Environmental and Occupational Health, University of Haifa School of Public Health, Haifa 3498838, Israel; claralewlev@gmail.com; 2Department of Occupational Medicine, Meuhedet Healthcare Organization, Tel Aviv 6777401, Israel

**Keywords:** air pollution, pediatric health, coal energy, power plants, particulates

## Abstract

Coal-based energy production is the most utilized method of electricity production worldwide and releases the highest concentration of gaseous, particulate, and metallic pollutants. This article aims to systematically review the public health impact of coal-fired power plant emissions on children’s health. PubMed, Web of Science, and Toxline databases were queried for the past 20 years. Inclusion criteria included original scientific articles with (a) coal-fired power plant exposure assessment, (b) at least one primary pediatric health outcome, and (c) assessment of potential sources of confounding and bias. Only morbidity and mortality studies were included; economic analysis and risk assessment studies without a primary health outcome were not included. Of 513 articles initially retrieved, 17 epidemiological articles were included in the final systematic review after screening and eligibility. The articles reviewed showed a statistically significant adverse effect on pediatric neurodevelopment; birth weight and pediatric respiratory morbidity was associated with exposure to coal-fired power plant emissions, primarily particulate matter and polyaromatic hydrocarbon exposure. There is a lack of consistency of exposure assessment and inadequate control of significant potential confounders such as social economic status. Future research should focus on improving exposure assessment models with an emphasis on source-apportionment and geographic information system methods to model power plant-specific emissions.

## 1. Introduction

The combustion of coal produces an exothermic reaction releasing particulate, gaseous, and metallic pollutants into the environment. Coal-fired power plants produce electricity through the rotation of a turbine by the steam produced when coal combustion occurs under high-pressure. A result of this combustion releases 84 of the 187 compounds listed as “hazardous air pollutants” by the U.S. Environmental Protection Agency [1]. Coal is the largest fuel source for electricity production worldwide and coal-fired power plants constitute a large majority of all emissions related to energy production. In the United States, coal-fired power plants account for 60% of all sulfur dioxide, 50% of mercury, 60% of arsenic, and 13% of nitrogen oxide emissions. A total of 75% of power generation in China is reliant on coal, accounting for nearly 50% of total SO_2_, 27% of NO_x_, and 11% of PM_10_ emissions in China [2]. Additionally, coal emissions from power plants represent the number one anthropogenic source of green-house gases worldwide. In the United States, approximately 81% of all greenhouse gas emissions are due to coal-fired power plant emissions, primarily carbon dioxide and to a lesser extent, nitrous oxides [1].

The health effects of coal on children are primarily known from the extensive research on indoor air pollution secondary to the combustion of coal for cooking and heating throughout the developing world. An IARC (International Agency for Research on Cancer) systematic review has determined that indoor emissions from the combustion of coal represent a group 1 carcinogen [3]. While indoor coal combustion continues to be a primary source of indoor air pollution for children in the developing world [4], coal-based energy production is a major contributor to ambient air pollution world-wide. Despite this, the majority of epidemiological literature on pediatric health effects from coal deals with indoor exposures and does not adequately address the potential public health risk associated with coal-based energy production.

The adverse impact of ambient air pollution on children’s health is unique due to their developing physiology, anatomy, metabolism, and health behaviors. Pediatric populations have unique exposures and an increased internal dose from air pollution due to their anabolic physiologic state, increased ventilation rate, and greater surface to body mass ratio when compared to adults. Children are uniquely vulnerable due to their developing neurological system and metabolism. Additionally, children are politically vulnerable, being unable to recognize environmental hazards or advocate for improved environmental health conditions. The health effects of ambient air pollution in pediatric populations has been widely studied and reviewed, although not specifically dealing with coal [5]. Recent studies have shown an increased risk of adverse respiratory health outcome from PM_2.5_ exposure [6], and the impact of emissions from oil refineries on lung function reduction among children [7]. In addition to the well-documented respiratory health impact, ambient air pollution is associated with adverse birth outcomes [8]. This article aims to systematically review the literature on the impact that air pollution from coal-fired power plants has on the morbidity and mortality of children worldwide. We present a critical assessment of the current literature, identify methodological limitations and impacts on policy, and suggest directions for future research.

## 2. Materials and Methods

### 2.1. Search Strategy and Inclusion Criteria

A systematic review of the epidemiological literature was conducted on PubMed, Web of Science, and Toxline (unique from PubMed) platforms utilizing an identification, screening, eligibility, and inclusion algorithm. Multiple search terms where used on all three platforms. All search terms included search restriction with the terms “children” and/or “pediatric” in the subject search. Search restrictions included English language, years of publication 1998–2018, and human participants.

In order to widen the scope and to minimize the likelihood of missing relevant literature, the term “power plant” was used in addition to “coal-fired”. In total, there were 684 unique search retrievals for each search term for all three databases (Table 1). In addition to articles identified through the search platforms, a citation review of the above listed terms added another 65 articles. Following the initial identification process, duplicate articles from different search terms were removed prior to starting the screening process. Figure 1 outlines a flow chart of the selection process adapted from the PRISMA group statement [9].

### 2.2. Eligibility Process

The abstracts of the 513 articles that were included in the different search results listed above were reviewed and narrowed the pool of relevant articles based on a narrow set of criteria:

Original scientific articles (not review articles);

Predatory journals were excluded based on guidelines published by Laine et al. [10];Coal-fired power plants represented one of the primary sources of exposures assessed;At least one primary dependent variable is a health outcome in children;Only morbidity and mortality studies were included. Economic analysis and exposure assessment studies without a health outcome were not included.

Following the screening for eligibility, 53 articles remained. Articles were included in the final critical review if statistical methods were based on accepted parameters of *p* < 0.05 and a power of >80% to minimize the risk of type I or type II error. A systematic effort was made to control for potential confounders, and both bias and misclassification were assessed, with attempts made to minimize their effects. Sixteen items were included in the final systematic review. A review protocol and article abstract instrument was developed and applied to all articles included in the final systematic review. Measures of effect (odds ratio, relative risk, hazard ratio, etc.) were tabulated and summarized in a structured format.

## 3. Results

Below, we summarize the methods, primary findings, and limitations of the epidemiological literature of the health impact of coal-fired power plant emissions on children. Table 2 summarizes the study population, exposure, and outcome assessment methods. Table 3 summarizes the primary measurement of effect for all studies reviewed.

### 3.1. Respiratory Outcomes

A spatial cross-sectional study of children aged 2 to 14 in Alberta, Canada, assessed the association between residential proximity to a coal-fired power plant and pediatric emergency room visits for asthma [11]. Exposure assessment was based on the postal code median distance to the coal-fired power plant without modeling for emissions. A spatial analysis of disease clustering showed an inverse association with distance from the power plant and asthma visits for children in the study area after adjusting for socioeconomic status, age, and gender. A statistically significant association was not seen when assessing proximity to a nearby petrochemical plant. The study did benefit from the use of a large population-based administrative dataset, which limited outcome misclassification and bias. The primary limitation of the study is the relatively crude assessment of exposure which was subject to misclassification and does not account for the plume effect of plant emissions.

A study of 196 children from 26 villages in Thailand surrounding a coal-fired power plant assessed the association between daily SO_2_ concentrations and the incidence of respiratory symptoms [12] in a time-series analysis. No significant association was seen between daily concentrations and respiratory symptoms in either asthmatic or non-asthmatic children in the cohort. One of the primary limitations of the study was the lack of personal exposure estimates and the reliance on only three ambient air monitors across a large area with diverse exposure pathways. The authors included PM_2.5_ and PM_10_ concentration in two-pollutant models; however, there was significant error in the exposure data which may have biased the results towards the null.

Children living in three communities near a major coal-fired power plant in Hadera, Israel, exhibited a significant rise in asthma and respiratory-related conditions, as well as reduced PFT in the decade after the plant became operational [13]. Exposure was assigned by measuring defined “air pollution ‘events’ in which the half-hourly averages for SO_2_ and NO_x_ were above an arbitrary threshold”, as measured by 12 regional air pollution monitors. Asthma prevalence increased from approximately 8% prior to the plants opening to 13% a decade later. Wheezing and shortness of breath increased from approximately 11% of children to 16%. After controlling for potential confounders, a significant association was found for asthma (OR: 1.79 [95% CI: 1.16, 2.74]) and wheezing/shortness of breath (OR: 1.59 [95% CI: 1.11, 2.28]). The primary limitation in this study was the lack of available exposure data and the crude method for assessing exposure. The “event method” of exposure assessment described in the article is prone to exposure misclassification.

Further cohort studies assessed respiratory outcomes among children living near the same coal-fired power plant in Hadera. Estimated personal exposure to NO_x_ and SO_2_ among 1181 children based on home address was compared with respiratory health status and pulmonary function tests [14]. The authors estimated exposure to “air pollution events” above an arbitrary concentration cut off as estimated from regional monitoring stations, identifying low, medium, and high exposure areas. Pulmonary function tests were lowest in the “high pollution” areas. Children in “high pollution” areas with a report of daily respiratory symptoms had larger decreases in pulmonary function than children without respiratory symptoms. This suggests that children with chronic respiratory disease may be more susceptible to respiratory effects associated with living in areas with higher emissions from power plants. Dubnov et al. [15] analyzed the same data for 1492 children residing near the Hadera, Israel power plant and found a significantly negative association between increasing exposure to NO_x_ and SO_2_ with decreasing forced expiratory volume in 1 second (FEV1) and forced vital capacity (FVC).

Peled and colleagues [16] conducted a time-series analysis comparing daily PM_10_ and PM_2.5_ concentrations with hospitalization and emergency room visits for children (0–3 years) residing near a major coal-fired power plant Ashkelon, Israel. Of the 3600 ER visits and 1134 hospital admissions, the highest rate of hospitalization was found in the city closest to the power plant, with a significant positive correlation between PM_2.5_ concentration and both respiratory ER visits (*p* = 0.02) and hospitalization (*p* = 0.03). A survey of daily respiratory symptoms and peek expiratory flow (PEF) was conducted for 285 asthmatic children living near the same coal-fired power plant in Ashkelon, Israel. Maximum daily PM_2.5_ concentrations were inversely associated with PEF for asthmatic children living in Ashdod, just north of the power plant (coefficient = −2.74; *p* < 0.001). While the study did show that elevated air pollution concentrations were associated with increased respiratory symptoms and decreased pulmonary function, the lack of exposure modelling estimating personal exposure limits the ability to attribute the observed adverse health effects to power plant emissions.

### 3.2. Birth Outcomes

The association between the distance of the residence to coal-fired power plants and adverse birth outcomes was assessed for 423,719 births from 2004 to 2005 in Florida, USA [17]. The authors also assessed prenatal exposure to PM_2.5_ based on the US Environmental Protection Agency Hierarchical Bayesian Prediction Model utilizing air monitor and national emission inventory data. Prenatal exposure to PM_2.5_ was significantly higher for children in close proximity to coal power plants (10.7 μg/m^3^) when compared to natural gas (9.5 μg/m^3^) and nuclear power plants (7.7 μg/m^3^). Infants born within 20 km of more than one coal-fired power plant had significantly higher odds of a low birth weight (OR: 1.12 [95% CI: 1.03, 1.22]), preterm delivery (OR: 1.20 [95% CI: 1.14, 1.25]), and very preterm delivery (OR: 1.23 [95% CI: 1.10, 1.36]). Significant associations were not seen with residential proximity to non-coal-fired power plants. This study only looked at residential proximity and did not estimate actual personal exposure to emissions. Although the study did not account for potential residential mobility during pregnancy, it did meaningfully control for socio-economic confounders.

Yang et al. [18] assessed birth weight among all children born within 30 miles of a major US coal-fired power plant in Pennsylvania between 1990 to 2006. They reported that mothers living within four counties designated as “down-wind” from the power plant during the last month of pregnancy were 0.4% to 6.5% more likely to have a lower birth weight infant, while a very low birth weight (VLBW) was 0.19% to 17.12% more likely. The study is subject to exposure misclassification due to the relatively broad exposure assessment based on wind direction and proximity to the power plant.

Mohorovic and colleagues [19] took advantage of a natural experiment when a large coal-fired power plant in Labin, Croatia, temporarily went off line for six months. The researchers tested methemoglobin levels as a biomarker of oxidative stress among pregnant woman three times during power plant operation and three times during the off season. Ground-level SO_2_ concentrations were also compared during the two study periods. A positive correlation was found between methemoglobin concentration and daily SO_2_ concentration while the power plant was operational. There was a gradual decline in methemoglobin concentration during the period when the power plant was un-operational (r = −0.60, *p* = 0.05); after the plant restarted operation, the average methemoglobin concentration increased (r = 0.73, *p* = 0.01).

A follow-up study [20] of the same cohort of pregnant women reported that frequencies of stillbirth and miscarriage were 60% lower during the “control” period when the power plant was un-operational (*p* = 0.03). The researchers hypothesized that emissions from the coal-fired power plant cause hemoglobin oxidation, resulting in elevated methemoglobin throughout the pregnancy, thereby resulting in fetal hypoxia and ultimately sudden fetal death.

### 3.3. Neurodevelopmental Outcomes

The Tongliang power plant in China provided a unique natural experiment as it operated seasonally until it was completely closed in 2004. Prior to the power plant’s closure in 2004, maternal polyaromatic hydrocarbon–DNA (PAH-DNA) adduct levels above the median adduct level were associated with a decreased birth head circumference (*p* = 0.057) and significantly (*p* < 0.05) reduced children’s weight at 18 months, 24 months, and 30 months. The study, however, did not directly estimate the contribution of coal-fired power plant PAH emissions [21]. Following the power plant closure, a prospective follow-up was conducted of the same cohort of 133 maternal-infant pairs who lived within 2.5 km of the power plant prior to its closure [22]. The researchers measured PAH-DNA adducts, lead and mercury in umbilical cord blood, and subsequently assessed neurodevelopment at age 2 using the Gesell Developmental Schedules. The authors reported an inverse association between the cord PAH-DNA adduct and motor, language, and total average developmental quotients (DQ). Cord lead was inversely associated with social and average DQ, while a 0.1 increase in cord adduct was associated with increased developmental delay (OR = 1.91 [95% CI: 1.22, 2.97]).

In a follow-up study, the neurodevelopmental benefits of reducing prenatal exposure to PAH following the closure of the Tongliang coal-fired power plant was assessed [23]. The study included two identical prospective cohort studies both preceding and following the closure of a major coal fired power plant in Tongliang, Chongqing, China. The authors reported that there were no longer significant associations between PAH-DNA adduct and developmental delay in the second post-closure cohort. The study concluded that “an intervention to eliminate emissions from a polluting coal-burning power plant was effective in improving developmental outcomes among children living” near a power plant. The authors reported that lower levels of the PAH-DNA adduct, higher concentrations of mature brain derived neurotrophic factor (BDNF) protein, and higher DQ scores were seen in the post power plant closure cohort when compared to the pre-closure cohort from three years earlier. Adduct concentrations were inversely associated with BDNF with motor, adaptive, and average DQ. The exposed pre-closure cohort showed a statistically significant association between B[a]P-DNA adducts and decreased head circumference [24]. This association disappeared in the post-closure cohort. Birth weight and height also increased in the post-closure cohort and were correlated with a decreased PAH-DNA concentration [25].

The strengths of the Perera and Tang studies are the prospective cohort design, use of molecular markers as a metric of physiologically significant internal dose, and adequate control for a number of potential confounders. The authors took advantage of a natural experiment to assess the reversal of the effect on child neurodevelopment which followed the plant’s closure. The main limitation in these studies, however, is that the authors did not model power plant-specific emissions. However, the assumption that the majority of the PAH exposure was due to the power plant is reasonable since there was a 1.5–3.5 times increase in ambient PAH concentration during the plant’s seasonal operation. However, some proportion of the reported detrimental neurodevelopmental affect from PAH exposure was potentially due to other industrial or transportation sources.

### 3.4. Cancer

Researchers in India analysed the concentration of PAHs in residential soils in an area surrounding a major coal-fired power plant [26]. The measured concentrations and soil ingestion pathway were utilized to estimate the lifetime average daily dose. Incremental life time cancer risk due to PAHs through soil ingestion was 1.5 × 10^−7^ for children. The authors premised that PAHs in soil samples originated from petrogenic and mixed pyrogenic activities such as coal combustion based on composition profiles and molecular ratios of PAHs in soils. The primary limitation of this study is solely relying on the ingestion pathway of soil and not including agricultural or water ingestion, as well as inhalation and dermal exposure, pathways.

### 3.5. Fluorosis

The phenomenon known as “coal-fired-pollution-induced fluorosis” is potentially due to increased burning of fluoride-rich coal and the subsequent deposition in soil, as well as locally grown produce [27], causing dental fluorosis. In order to further investigate this hypothesis, the concentrations of 11 metals in soil near a coal-fired power plant in Chongqing, China, were measured. The whole blood concentration of five metals and urinary fluoride levels among fluorosis cases and controls were measured. Ni, I, F, and Hg concentrations and soil pH values were positively correlated with fluorosis prevalence. Cu, Zn, Mg, and Fe levels of the children with fluorosis were lower and urine fluoride levels were higher when compared to children without fluorosis. This suggests that children living near coal-fired power plants in China have a higher risk of fluorosis due to the increased internal dose of fluoride while lacking exposure to some anti-fluoride elements [28].

## 4. Discussion

This review illustrates the significant impact on pediatric respiratory health, neurodevelopment, and birth outcomes from gaseous, particulate, and metal coal-fired power plant emissions. Intervention studies such as those conducted following the closure of the Tongliang power plant in China [21,22,23,24,25] and the opening of a coal-fired power plant in Israel [13,14,15,16], provide a natural experiment to study the impact of ambient emissions from coal combustion during energy production. These studies have a number of benefits. By using the same exposed population of children, residual confounding is limited. The cohort provides its own control for comparison pre and post intervention. This assumes no major migration or demographic change in the intervening period. This is only beneficial for more acute health outcomes as chronic conditions such as cancer rates are not likely to change significantly immediately following power plant closure or conversion, possibly explaining the lack of studies of cancer in children.

An additional benefit of intervention studies and natural experiments is that they help to strengthen a causal relationship between coal emissions and increased mortality and morbidity based on reversal of the effect component of Hill’s criteria [29]. Intervention studies are also useful in determining the benefit of plant closure and conversion for cost-benefit analyses, the impact of which plays heavily on policy considerations.

There is considerable inconsistency in exposure assessment and modeling in the literature reviewed. The assumptions and parameters used in exposure models have a significant impact on the studies’ outcomes. The error introduced by exposure misclassification varied significantly among the studies reviewed. Consequently, the internal validity varied greatly among the studies, primarily due to inconstancy in the exposure assessment. Consistency in the exposure assessment will greatly improve the state of the literature by increasing the internal and external validity of the literature.

Future research should focus on improving exposure assessment models, with an emphasis of source-apportionment and GIS methods to isolate power plant-specific emissions. The majority of studies reviewed did not assess power plant-specific emissions. There are a number of co-exposures that coincide with emissions from power production, including transportation and industrial sources of exposure. By not estimating power plant-specific emissions, researchers are limited in their ability to assess the impact of coal-based power generation on children’s health. This also limits the ability to use the epidemiological literature in cost-benefit analyses when considering the cost of plant closure or transitioning to a cleaner form of electricity production. The studies which do, however, include emission-based estimates of personal exposure were able to show the contribution that coal-specific pollution has on pediatric morbidity and mortality worldwide.

The literature reviewed does not address the long-term health effects on children from greenhouse gas emissions. With the rising awareness and increasing body of evidence linking climate change to the public’s health, it is imperative to understand the contribution of greenhouse gas emissions from coal-based power production on health.

The vast majority of studies reviewed inadequately controlled for social economic status. Children who live closest to coal-fired power plants are more likely to have lower SES. This presents a significant source of confounding as SES is highly associated with all of the outcomes investigated. SES may also be a significant effect modifier as poverty, decreased parental education, and decreased nutrition increase the detrimental effect that environmental pollution has on the health of children.

Multiple pollutant models were utilized in a few studies; however, the vast majority of studies focused on single pollutant measurements and models. Considering the large number of pollutants attributed to coal production, the interaction toxicants have with one another is important to assess to fully understand the impact that coal emissions have on public health.

A meta-analysis was not included in this review for a number of reasons. Multiple different exposures were assessed in the studies reviewed. Consequently, there were not enough studies that looked at one specific exposure (particulates, NO_x_, SO_x_, metals) to allow for a meta-analysis. As mentioned above, there is a lack of consistency in the exposure assessments utilized, again minimizing the number of studies with comparable results. Finally, the articles reviewed cover a range of health outcomes, from birth outcomes and cancer to neurological and respiratory conditions. Except for neurological conditions, there is no one health outcome in the current review with enough comparable data to allow for a meta-analysis.

The public health impact of coal-based energy production on children is a particularly timely issue as the general trend of the decreasing dependence on fossil fuel-based energy, specifically coal, has been called into question in the United States, the global leader in energy consumption [30]. While there has been a steady decline of coal-based energy production in the US over the past 30 years, political pressures since the inauguration of President Trump in 2017 have reopened the development and investment of coal-based energy and coal mining [31]. World-wide, over 1600 coal-fired power plants are either under construction or planned in 62 countries [32]. In most parts of the world, coal-fired power plants are the primary source of power generation and their capacity is on the rise. Consequently, the negative impact of coal emissions on child health will likely increase and continue to be a significant public health concern.

## 5. Conclusions

There is a growing body of epidemiological evidence associating emissions from coal-fired power plants with the increased morbidity of children in both developing and developed countries. The articles included in the systematic review show an overall statistically significant adverse effect on pediatric neurodevelopment, birth weight, and childhood respiratory morbidity associated with exposure to coal-fired power plant emissions, primarily particulate matter, and polyaromatic hydrocarbon exposure. Significant improvement in pediatric morbidity is associated with the closure of coal-fired power plants. There is a lack of consistency of exposure assessment and inadequate control of significant potential confounders such as social economic status. Future research should focus on improving exposure assessment models with an emphasis on source-apportionment and GIS methods to isolate power plant-specific emissions. Additionally, future research should emphasize the impact of greenhouse gas emissions and global warming on children’s health, focus on natural experiments following power plant closures, and emphasize the economic benefit and policy implications of ending the global dependence on coal-based energy production.

## Figures and Tables

**Figure 1 ijerph-16-02008-f001:**
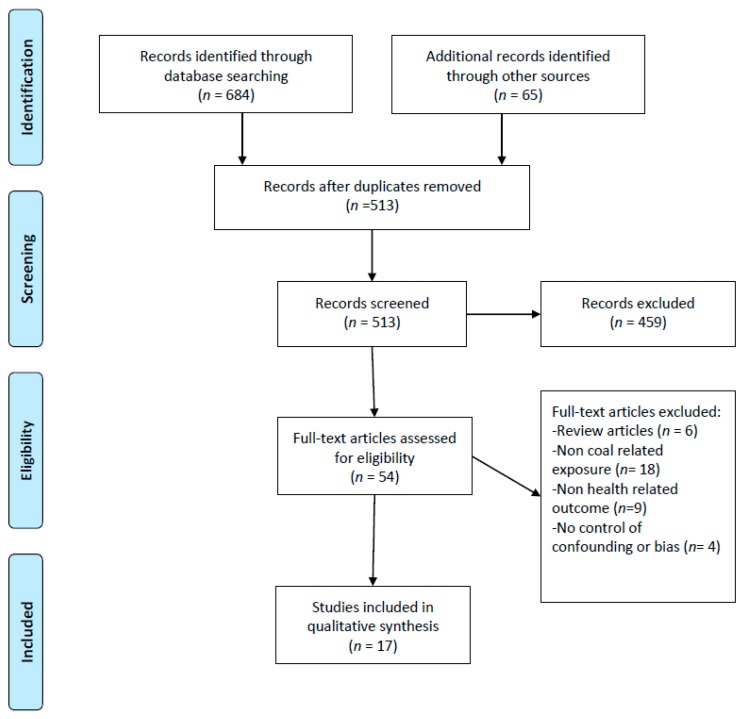
Flow chart of study selection, screening, and eligibility.

**Table 1 ijerph-16-02008-t001:** Search terms and query results.

Term	PubMed	Web Science	Toxline ^1^	Unique Results
coal-fired AND morbidity	18	9	5	22
coal-fired AND mortality	2	2	1	3
coal-fired AND health	31	54	9	64
power plant AND morbidity	129	13	11	132
power plant AND mortality	18	58	30	61
power plant AND health	192	353	111	402
**Total**	390	489	167	684

^1^ unique Toxline search without PubMed crossref.

**Table 2 ijerph-16-02008-t002:** Summary of study population, exposure, and outcome metric.

Author	Population	Exposure Metric	Health Outcomes
Rodriguez-Villamizar (2018)	Cross-sectional. Alberta, Canada	Residential proximity to CFPP	Pediatric ER asthma visits; record review
Aekplakorn (2003)	Pediatric time-series. Thailand	SO_2_ and PM concentrations from 3 monitoring stations	Self-report incidence respiratory symptoms; questionnaire data
Goren (1997)	Pediatric cohort. Hadera, Israel	Nox and SO_2_ peak “events”, 12 air monitoring stations	Change in PFT, respiratory symptom and asthma prevalence
Yogev-Baggio (2010)	Pediatric cohort. Hadera, Israel	Nox and SO_2_ peak “events” at 12 regional air monitoring stations	PFT, asthma diagnosis, respiratory sx; record review
Dubnov (2006)	Pediatric cohort. Hadera, Israel	Nox and SO_2_ peak “events”, 12 air monitoring stations	PFT, asthma diagnosis, respiratory sx; record review
Peled (2004)	Time-series analysis. Ashkelon, Israel	PM_10_ and PM_2.5_, 10 monitoring stations	ER and hospitalization, peek flow and respiratory symptoms in infants; record review
Ha (2015)	Retrospective birth cohort, Florida, USA	Prenatal PM_2.5_; air monitor stations, national emissions inventory estimated to residence	Preterm delivery, birthweight; State vital statics records.
Yang (2017)	Retrospective birth cohort. Pennsylvania, USA	Residential proximity to power plant (<20 km)	Low birth and very low birth weight; vital statistics records
Mohorovic (2003)	Prospective pregnancy cohort. Labin, Croatia	SO_2_ concentrations during plant operation and closure	Methemoglobin levels as a biomarker of oxidative stress
Mohorovic (2010)	Prospective pregnancy cohort. Labin, Croatia	SO_2_ concentrations during plant operation and closure	Incidence of stillbirth and miscarriage
Tang (2006)	Birth cohort. Chongqing, China	Cord blood: PAH-DNA adducts, lead and mercury	Birth head circumference, birth weight
Tang (2008)	Follow-up birth cohort. Chongqing, China	Cord blood: PAH-DNA adducts, lead and mercury	2-year GDS neurodevelopment
Tang (2014a)	Follow-up birth cohort. Chongqing, China	Cord blood: PAH-DNA adducts, lead and mercury	2-year GDS neurodevelopment and birth outcomes
Tang (2014b)	Follow-up birth cohort. Chongqing, China	Cord blood: PAH-DNA adducts	2-year GDS neurodevelopment and birth outcomes; mature BDNF protein
Perera (2008)	Follow-up birth cohort. Chongqing, China	Cord blood: PAH-DNA adducts, lead and mercury	2-year GDS neurodevelopment
Kumar (2014)	Risk assessment. Korba, India	PAH, estimated dose based on soil samples	Estimated life-time cancer risk
Yang (2017)	Retrospective birth cohor. Pennsylvania, USA	Residential proximity to power plant (<20 km)	Low birth and very low birth weight; vital statistics

Coal-fired power plant (CFPP), Emergency Room (ER), Particulate Matter (PM), Pulmonary Function Tests (PFT), Polyaromatic Hydrocarbons (PAH), Global Deterioration Scale (GDS).

**Table 3 ijerph-16-02008-t003:** Summary of studies’ primary effect measurement.

Author	Measurement of Effect
Rodriguez-Villamizar (2018)	Relative Risk for ER asthma visits 10.4 comparing children residing within “power plant area” vs. outside
Aekplakorn (2003)	No statistically significant association between self-report incidence respiratory symptoms and PM and SO_2_ concentrations
Goren (1997)	Asthma OR 1.79 [95% CI: 1.16, 2.74], wheezing + SOB: OR 1.59 [95% CI: 1.11, 2.28]
Yogev-Baggio (2010)	Largest FEV1 deficit associated with living in the “high” air pollution area versus “low” air pollution area is observed in the children with respiratory symptoms (19.6%)
Dubnov (2006)	Increasing exposure to NOx and SO2 with decreasing FEV1 and FVC
Peled (2004)	Positive correlation between PM_2.5_ concentration and both respiratory ER visits (*p* = 0.02) and hospitalization (*p* = 0.03)
Ha (2015)	Infants born within 20 km of more than one coal-fired power plant had significantly higher odds of a low birth weight (OR: 1.12 [95% CI: 1.03, 1.22]), preterm delivery (OR: 1.20 [95% CI: 1.14, 1.25]), and very preterm delivery (OR: 1.23 [95% CI: 1.10, 1.36]
Yang (2017)	Increase of 0.4% to 6.5% for low birth weight infant; Increase of 0.19% to 17.12% for very low birth weight (VLBW)
Mohorovic (2003)	Maternal methemoglobin concentration increased during plant operation (r = 0.73, *p* = 0.01)
Mohorovic (2010)	Stillbirth and miscarriage decreased 60% during period when power plant was nonoperational (*p* = 0.03)
Tang (2006)	PAH–DNA adduct levels above the median adduct level are associated with decreased birth head circumference (*p* = 0.057) and significantly (*p* < 0.05) reduced children’s weight at 18 months, 24 months, and 30 months
Tang (2008)	Inverse association cord PAH-DNA adduct and motor, language, and total average developmental quotients. Cord adduct associated with increased developmental delay OR: 1.91 [95% CI: 1.22, 2.97]
Tang (2014a)	Decreased PAH-DNA levels post closure associated with increased head circumference and
Tang (2014b)	PAH-DNA adducts were inversely associated with mBDNF, as well as scores for motor (*p* = 0.05), adaptive (*p* = 0.022), and average (*p* = 0.014) DQ
Perera (2008)	No statistically significant association between PAH-DNA adduct and developmental delay after power plant closure
Kumar (2014)	Incremental life time cancer risk due to PAHs through soil ingestion was 3.1 × 10^−7^ for adults and 1.5 × 10^−7^ for children
Yang (2017)	Increase of 0.4% to 6.5% for low birth weight infant; Increase of 0.19% to 17.12% for very low birth weight (VLBW)

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
