# Peer review of "Impact of Coal-fired Power Plant Emissions on Children’s Health: A Systematic Review of the Epidemiological Literature"

_ijerph, 2019, doi:10.3390/ijerph16112008_

Round 1

Reviewer 1 Report

Thank you for the opportunity to review this paper. This systematic review (n=16 studies) summarizes the relationship between coal-fired power plants and children health outcomes including perinatal health, respiratory outcomes, cancer, neurodevelopmental outcomes. Overall, the authors found that there is an overall association between power plant emissions and pediatric neurodevelopment, birthweight, and respiratory illnesses, particularly with fine particulate matter and polyaromatic hydrocarbons. Below are my specific comments:

Abstract

-Line 21-The authors mentioned that "there is an overall statistically significant adverse effect on pediatric neurodevelopment...". This implies that they actually performed a meta-analysis. If this is true, I would also mention the statistical methods used in the analyses. If meta-analysis was not done, I would rephrase.

Intro

-Page 2, line 5: define IARC

-The paper is focused on children, I highly suggest that the authors discuss more about the rationale to focus on this population (e.g, vulnerability). 

Methods

--Search term: I am a little worried that the current search terms could miss some studies that use specific terms (e.g., cancer, etc) instead of 'morbidity' or 'mortality'.  For example, if an article uses 'power plant' and 'adverse pregnancy outcomes' as key terms, it would have been missed?

--Page 3, lines 12-13: I am not sure if I follow. Does this mean that only articles with significant results were included?

Page 3, lines 12-18: It sounds like the authors are performing a meta-analysis. In this case, I think a statistical analysis section is necessary. In addition, clarification on the method used to access qualitative quality of the studies is also critical. In other words, which qualitative assessment method did the authors use? (a commonly used method is the Newcastle-Ottawa Scale). On the other hand, if meta-analysis is not done, please justify (may be not enough studies with the same combination of exposure/outcome?). Currently the paper is set up as if it's going to be a systematic review/meta-analysis.

Results

--Table 2: I also suggest including the sample size in this Table. In addition, methods of outcome assessment should also be added (e.g. medical records? personal interview?). Definitions for abbreviations are also recommended.

- Page 5-- Reference 17 (Florida study) is not in the Table.

-Table 3 is currently dependent on Table 2, which makes some of the data not stand-alone. For example, for Aekplakorn (2003), we see 'no statistically significant association'. This is not very helpful unless we read Table 2 at the same time. I would suggest combine the 2 tables if possible. If space does not allow, may be specify no statistically significant associations between what?

-

Conclusion:

-Line 34: the authors mentioned that there is an overall statistically significant adverse effect on pediatric neurodevelopment...etc. However, there is no meta-analysis for a summary estimate. I would reword.

Author Response

Thank you again for your consideration of our manuscript for the upcoming series entitled, “Children, Air Pollution, and the Outdoor Urban Environment”.  Below is a complete, point by point response on how we addressed the issues raised by the reviewers.  We very much appreciate the reviewer’s comments.  Please let us know if there are any further comments that we can address at this point. 

Reviewer 1 

Abstract 

-Line 21-The authors mentioned that "there is an overall statistically significant adverse effect on pediatric neurodevelopment...". This implies that they actually performed a meta-analysis. If this is true, I would also mention the statistical methods used in the analyses. If meta-analysis was not done, I would rephrase. 

The article is a systematic review with a summary of key findings, and critical appraisal of methodological limits and public health significance of the published literature.  The article is not a meta-analysis and this will be clarified throughout the article.  Mention of statistical significance is referring to the key conclusions of the articles having statistical significance.  The sentence has been rephrased to state that “The articles reviewed showed an overall statistically significant adverse effect....”.  This point has been clarified in other points throughout the manuscript as well. 

-Page 2, line 5: define IARC 

Included 

-The paper is focused on children, I highly suggest that the authors discuss more about the rationale to focus on this population (e.g, vulnerability).  

A discussion of the unique exposures and vulnerability of children is included in the introduction including references from recent peer- reviewed journals.  We have also noted that children are politically vulnerable, being unable to recognize environmental hazards or advocate for improved environmental conditions. 

Methods 

--Search term: I am a little worried that the current search terms could miss some studies that use specific terms (e.g., cancer, etc) instead of 'morbidity' or 'mortality'.  For example, if an article uses 'power plant' and 'adverse pregnancy outcomes' as key terms, it would have been missed? 

Searches were conducted again including specific health outcomes instead of “morbidity”. Specific health outcomes were selected based on biological plausibility. We used the same search protocol as stated in the methodology section including “cancer”, “neurodevelopment”, “birth weight”, “pregnancy outcomes”, “asthma”, “respiratory conditions”, “bronchitis”, “pneumonia”.  There were no additional articles identified after the identification and screening stages of the protocol.  In conclusion, we believe it is unlikely that further articles were missed from what is presented in our manuscript.   

--Page 3, lines 12-13: I am not sure if I follow. Does this mean that only articles with significant results were included? 

All articles were included in the systematic review including those that were not significantly significant.  In line 12-13 we note that only articles that had acceptable statistical criteria were included.  For example, articles which did not have statistical power beyond 80% were not included as there is an increased risk of type II error.  Multiple articles are included in the final review which do not have statistically significant results (PereraAekplakorn studies), however statistical analysis was adequate and included accepted statistical criteria. This point is clarified in the manuscript text.  

- Page 3, lines 12-18: It sounds like the authors are performing a meta-analysis. In this case, I think a statistical analysis section is necessary. In addition, clarification on the method used to access qualitative quality of the studies is also critical. In other words, which qualitative assessment method did the authors use? (a commonly used method is the Newcastle-Ottawa Scale). On the other hand, if meta-analysis is not done, please justify (may be not enough studies with the same combination of exposure/outcome?). Currently the paper is set up as if it's going to be a systematic review/meta-analysis. 

Thank you for this comment.  A meta-analysis was not included for this very point: there were not enough studies with consistent exposure assessment or outcome classification in order to perform a meta-analysis.  While the Tang studies did all look at neurodevelopmental outcomes associated with coal power plant exposures, there were only three studies and they looked at different periods of exposure.  We have made it clearer in the manuscript that the article is not a meta-analysis and expound upon the reasons why a meta-analysis was not included in the discussion section.   

Results 

--Table 2: I also suggest including the sample size in this Table. In addition, methods of outcome assessment should also be added (e.g. medical records? personal interview?). Definitions for abbreviations are also recommended.   

Definitions for abbreviations have been added for both tables.  Suggestion to add outcome assessment methods is well received.  This has been added to the “Health outcomes” column in table 2.  Sample size was not added as it is not critical to the summary of the findings and would risk overcrowding the table as each study included a different n for different sub-analysis.  

- Page 5-- Reference 17 (Florida study) is not in the Table. 

Thank you.  Added.   

-Table 3 is currently dependent on Table 2, which makes some of the data not stand-alone. For example, for Aekplakorn (2003), we see 'no statistically significant association'. This is not very helpful unless we read Table 2 at the same time. I would suggest combine the 2 tables if possible. If space does not allow, may be specify no statistically significant associations between what? 

Space and formatting limitations do not allow combining the two tables.  We will describe in Table 3 what specific associations are being referred to.  For example, instead of “no statistically significant association”, the table reads” no significant association between PM and respiratory symptoms”.   

Conclusion:-Line 34: the authors mentioned that there is an overall statistically significant adverse effect on pediatric neurodevelopment...etc. However, there is no meta-analysis for a summary estimate. I would reword. 

This has been reworded to make clear that the review is not a meta-analysis. 

Reviewer 2 Report

This is an extremely well written systematic review which explores the impact of air pollution generated from coal-fired power plants and children's health outcomes. The manuscript follows the PRISMA guidelines well. I find it difficult to fault what the authors have done.

Author Response

No changes, additions or edits were requested by this reviewer.  

Reviewer 3 Report

The research was thoroughly carried out with application of clear criteria for the dataset selection and are up-to state-of art. The results are significant from environmental and health point of view. I have a few minor comments:

-   I think that the diagram (Figure 1) should contain key words / terms after which the authors chose publications / papers.

-    I believe that Introduction should be extended. Above all, the topic should be better and more widely justified.

-   The works selected in the review collected in Tables 2 and 3 should be cited in alphabetical order or chronologically. Besides, they should be properly cited.

-   Conclusions should contain much wider recommendations for further research in the selected topic.

-  The work should be improved in terms of editing (indexes, conversion 0 to 0, etc.).

Author Response

Thank you again for your consideration of our manuscript for the upcoming series entitled, “Children, Air Pollution, and the Outdoor Urban Environment”.  Below is a complete, point by point response on how we addressed the issues raised by the reviewers.  We very much appreciate the reviewer’s comments.  Please let us know if there are any further comments that we could address at this point.   

Reviewer 3: 

-   I think that the diagram (Figure 1) should contain key words / terms after which the authors chose publications / papers. 

The key word terms used is described in the methods section “2.1. Search strategy”.  In this section the different key word terms are outlined including the number of retrievals for each term across the different search engines.  Table 1 does list all the key word terms and the specific number of papers in the query results.  The number of unique results for each search term and total number for each database is listed.  

-    I believe that Introduction should be extended. Above all, the topic should be better and more widely justified. 

Thank you.  The introduction has been extended to include a more in depth overview of pediatric environmental health and the unique vulnerabilities of children.  The discussion of the public health impact, economic and political considerations of coal based energy is also expanded upon in the introduction.  

-   The works selected in the review collected in Tables 2 and 3 should be cited in alphabetical order or chronologically. Besides, they should be properly cited. 

The works are cited in chronological order as they appear in the result section of the manuscript. As such, the articles are grouped based on specific health outcome.  This enables the reader to compare outcomes of studies with similar health outcomes (respiratory outcomes, neurodevelopmental outcomes).  We believe this is more functional and meaningful in comparing articles in the systematic review than an alphabetical ordering or by publication date.   If the reviewers would still prefer an alphabetical or chronological order, we would be happy to reorder the tables.  

-   Conclusions should contain much wider recommendations for further research in the selected topic. 

The discussion section has a much more in depth discussion of recommendations for further research.  This includes detailed recommendations on exposure assessment modeling, control for socioeconomic confounding, multi-pollutant models, evaluation of chronic health conditions and including climate change as an independent exposure.  The conclusion section contains a more concise summary of the much wider recommendations included in the discussion section (Page 8, line 18 until Page 9, line 37).  

-  The work should be improved in terms of editing (indexes, conversion 0 to 0, etc.). 

The authors have reviewed the manuscript carefully and have sent it for external editing to correct any grammatical, spelling or syntax errors.  I am not clear what conversion 0 to 0 is referring to.   

Round 2

Reviewer 1 Report

The authors have sufficiently addressed my concerns. I have no additional comment.